# A Multigene-Panel Study Identifies Single Nucleotide Polymorphisms Associated with Prostate Cancer Risk

**DOI:** 10.3390/ijms24087594

**Published:** 2023-04-20

**Authors:** Maria Antonietta Manca, Fabio Scarpa, Davide Cossu, Elena Rita Simula, Daria Sanna, Stefano Ruberto, Marta Noli, Hajra Ashraf, Tatiana Solinas, Massimo Madonia, Roberto Cusano, Leonardo A. Sechi

**Affiliations:** 1Dipartimento di Scienze Biomediche, University of Sassari, 07100 Sassari, Italy; 2Dipartimento di Scienze Mediche, Chirurgiche e Sperimentali, Università di Sassari, 07100 Sassari, Italy; 3Struttura Complessa di Urologia, Azienda Ospedaliera Universitaria, 07100 Sassari, Italy; 4CRS4, 09010 Pula, Italy; robycuso@crs4.it; 5Struttura Complessa di Microbiologia e Virologia, Azienda Ospedaliera Universitaria, 07100 Sassari, Italy

**Keywords:** *IL2RA* and *TNFRSF1B*, Gleason scores, *SLC11A1*, TNFRSF1B, PSA, prostate cancer

## Abstract

The immune system plays a critical role in modulating cancer development and progression. Polymorphisms in key genes involved in immune responses are known to affect susceptibility to cancer. Here, we analyzed 35 genes to evaluate the association between variants of genes involved in immune responses and prostate cancer risk. Thirty-five genes were analyzed in 47 patients with prostate cancer and 43 healthy controls using next-generation sequencing. Allelic and genotype frequencies were calculated in both cohorts, and a generalized linear mixed model was applied to test the relationship between prostate cancer risk and nucleotide substitution. Odds ratios were calculated to describe the association between each single nucleotide polymorphism (SNP) and prostate cancer risk. Significant changes in allelic and genotypic distributions were observed for *IL4R*, *IL12RB1*, *IL12RB2*, *IL6*, *TMPRSS2*, and *ACE2*. Furthermore, a generalized linear mixed model identified statistically significant associations between prostate cancer risk and SNPs in *IL12RB2*, *IL13*, *IL17A*, *IL4R*, *MAPT*, and *TFNRS1B*. Finally, a statistically significant association was observed between *IL2RA* and *TNFRSF1B* and Gleason scores, and between *SLC11A1*, *TNFRSF1B* and PSA values. We identified SNPs in inflammation and two prostate cancer-associated genes. Our results provide new insights into the immunogenetic landscape of prostate cancer and the impact that SNPs on immune genes may have on affecting the susceptibility to prostate cancer.

## 1. Introduction

Prostate cancer (PCa) is one of the most widespread noncutaneous cancers in men. The incidence of PCa appears to be affected by several risk factors, among which inflammation plays a prominent role [1,2]. Inflammation appears to play a special role in promoting the onset and progression of tumors by enabling an immunosuppressive tumor environment [3]. Inflammation is the physiological response of the body to deleterious stimuli such as pathogens and tissue damage, which results in tissue repair. In some conditions, dysregulation of this finely regulated process can result in a chronic inflammatory state, which, in turn, can sustain tumorigenesis. Cancer cells originate from the accumulation of genetic alterations, such as single nucleotide polymorphisms (SNPs), copy number variation, chromosomal rearrangements, loss of heterozygosity, and allelic imbalance, which result in dysregulation of cellular homeostasis and consequent expansion. Furthermore, cancer cells develop several systems to evade immune surveillance, such as loss of antigenicity, programmed death-ligand 1, and limited access to the primary tumor through the extracellular matrix. Furthermore, the immunological state of the tumor microenvironment (TME) is crucial for the implementation of cancer immunotherapy [4].

Advances in next-generation sequencing (NGS) technologies have enabled the simultaneous sequencing of millions of nucleotides and investigation of unique sets of genes for diagnostic, prognostic, therapeutic, and research purposes. Many studies have investigated the association between polymorphisms in immune system-related genes and cancer risk [5,6,7,8,9,10]. To this end, we analyzed genomic DNA derived from the peripheral blood mononuclear cells (PBMCs) of 47 patients diagnosed with PCa and 43 healthy controls (HCs). The MiSeq platform (Illumina) was used to analyze a customized gene panel of 35 genes (*ACE*, *ACE2*, *ATG7*, *CD28*, *IFNG*, *IFNGR1*, *IFNGR2*, *IL12A*, *IL12B*, *IL12RB1*, *IL12RB2*, *IL13*, *IL17A*, *IL18*, *IL18R1*, *IL2*, *IL2RA*, *IL4*, *IL4R*, *IL5*, *IL6*, *IL7R*, *IRF5*, *LAG3*, *MAPT*, *NR1I2*, *PACGR*, *PARK2*, *PTPN22*, *SCLC11A1*, *TMPRSS2*, *TNFRS*, *TNFSF1B*, *TRAF6,* and *VDR*). We then evaluated the possible association between the identified significant genetic polymorphisms and the clinical parameters of patients with PCa.

## 2. Results 

### 2.1. Allele and Genotype Frequency Distribution in PCa Patients and HCs

The present study analyzed a total of 47 patients with PCa and 43 HCs using NGS, and comprehensive gene profiling was conducted on a panel of 35 genes. Table 1 summarizes the distribution of significant allele and genotype frequencies between PCa patients and HCs. Among the nucleotide substitutions analyzed, seven SNPs, located within *IL4R*, *IL12RB1*, *IL12RB2*, *IL6*, *TMPRSS2*, and *ACE2* were found associated with PCa risk. Both the C allele and the CC genotype of *IL4R* rs2301807 were inversely associated with PCa risk (OR: 0.007, CI: 0.000–0.118, *p* < 0.0001; OR: 0.014, CI: 0.001–0.244, *p* < 0.0001, respectively). The AA genotype in the 3′ UTR of *IL12RB1* rs3746190, along with the C allele of *IL12RB2* rs2307145, were found associated with a lower risk of PCa (OR: 0.062, CI: 0.003–1.243, *p* = 0.013; OR: 0.194, CI: 0.039–0.971, *p* = 0.038, respectively). The G allele of the intronic variant *IL6* rs2069832 was positively associated with a higher risk of PCa (OR: 2.583, CI: 1.092–6.114, *p* = 0.028). In addition, both the A allele and CA heterozygous genotype of the intronic variant *TMPRSS2* rs140141551 were both associated with a higher risk of PCa (OR: 7.535, CI: 0.922–61.586, *p* = 0.0382; OR: 8.205, CI: 0.922–61.586, *p* = 0.024, respectively). Conversely, the G allele and the AG genotype of *TMPRSS2* rs11701576 were found to be associated with a decreased risk of PCa (OR: 0.309, CI: 0.097–0.977, *p* = 0.037; OR: 0.263, CI: 0.077–0.899, *p* = 0.027, respectively). Finally, a synonymous variant was identified within the *ACE2* gene, and the T allele of rs35803318 was found to be associated with a higher risk of PCa (OR: 11.743, CI: 0.659–209.216, *p* = 0.022).

### 2.2. Association between Nucleotide Substitution for Each Locus Examined and PCa Estimation

Multivariate regression analysis was performed using backward selection to assess the relationship between nucleotide substitutions at a given locus and PCa. To test this relationship, a generalized linear mixed model (GLMM) was implemented. The first GLMM runs were applied individually to each gene to identify loci statistically associated with the occurrence of PCa. Statistical significance was set at *p* < 0.05, with a few exceptions for borderline values, which were retained to avoid premature selection in intermediate models (Table 2). Odds ratio (ORs) calculations were performed as well on statistically significant variables representing useful predictors (Table 3). Statistically significant variables identified were then used to build a new dataset, and GLMMs runs were applied.

Using the predictors identified in Table 3, the analysis was performed considering statistically significant variables together (Table 4). ORs were calculated to assess their association with PCa risk (Table 5). Positive associations with PCa risk were observed for *IL12RB2* rs2228420 (OR: 1.687, CI: 1.282–2.219, Pr(>|t|) = 0.000347), *IL13* rs20541 (OR: 1.850, CI: 1.369–2.500, Pr(>|t|) = 0.000138), *IL17A* rs7747909 (OR: 1.214, CI: 1.040–1.418, Pr(>|t|) = 0.015979), *MAPT* rs2258689 (OR; 1.243, CI: 1.052–1.468, Pr(>|t|) = 0.012260), *TNFRSF1B* rs2275416 (OR: 1.377, CI: 1.042–1.820, Pr(>|t|) = 0.027425). A negative association with PCa risk was instead observed for *IL12RB2* rs2229546 (OR: 0.606, CI: 0.477–0.822, Pr(>|t|) = 0.001876), *IL4R* rs2301807 (OR: 0.653, CI: 0.555–0.768, Pr(>|t|) = 1.87 × 10^−6^), and *TNFRSF1B* rs636964 (OR: 0.678, CI: 0.476–0.965, Pr(>|t|) = 0.034034). Additionally, no statistically significant association between PCa risk and the nucleotide substitutions found within *ACE*, *ATG7*, *CD28*, *IFNG*, *IFNGR1*, *IFNGR2*, *IL12A*, *IL12B*, *IL18*, *IL18R1*, *IL2*, *IL2RA*, *IL4R*, *IL5*, *IL7R*, *IRF5*, *LAG3*, *R1I2*, *PACGR*, *PTPN22*, *TNFRS*, *TRAF6*, and *VDR.*

Finally, GLMM analysis was performed to explore the relationship between statistically relevant variants identified as useful predictors in the initial model (Table 2) and the clinical characteristics of patients with PCa (Table 5). Particularly, the following clinical characteristics were used for the analysis: Gleason score (GS), serum prostate-specific antigen (PSA) levels, age, clinical stage (cTNM), and body mass index (BMI).

Two polymorphisms (*IL2RA* rs7093069 and *TNFRSF1B* rs2275416) were significantly associated with GS. Accordingly, the variables rs7093069 and rs2275416 (within *IL2RA* and *TNFRSF1B* genes, respectively) represent good predictors for the PCa severity. Indeed, with GS with values of around 6, rs7093069 and rs2275416 do not appear as informative variables, while with values of around 9, they appear strongly associated with the severity of the pathology. In addition, *SLC11A1* rs17221959 and both *TNFRSF1B* rs636964 and rs2275416 were found to be significantly associated with serum PSA levels. Overall, only GS and PSA have been positively selected by the model, while other variables (age, cTNM, and BMI) were removed during the backward selection. Indeed, in the dataset here, analyzed age, cTNM, and BMI demonstrated to have no association with PCa, nor alone or taken together.

## 3. Discussion

The importance of the immune system in cancer onset and progression has been well established in cancer research. In the last decade, the notion of engaging the immune system as a patient-personalized weapon against cancer cells has undoubtedly revolutionized cancer therapy. A growing body of evidence has investigated the presence of variants in genes implicated in anti-tumor immune responses and cancer risk. Therefore, we investigated the presence of polymorphisms in a panel of 35 genes involved in pro- and anti-inflammatory responses, autophagy-lysosome pathway and two PCa-associated genes (*ACE2* and *TMPRSS2*) in a cohort of patients with PCa and a cohort of HCs. We identified several novel polymorphisms that contributed to PCa susceptibility. In particular, we found that six SNPs in several key genes, namely *IL4R*, *IL12RB1*, *IL12RB2*, *TMPRSS2*, and *TNFRSF1B*, were associated with a lower risk of PCa. In contrast, the eight SNPs identified in *IL6*, *TMPRSS2*, *ACE2*, *IL12RB2*, *IL13*, *IL17A*, *MAPT,* and *TNFRSF1B* were associated with a higher risk of PCa. PCa patients with specific polymorphisms in T-helper cytokine genes (*IL13* rs20541 and *IL4R* rs2301807) may have an altered T-helper response and consequently, a potential imbalance in their immune system. Altered expression and increased circulating levels of both pro- and anti-inflammatory cytokines have been reported in several types of cancer [11]. Particularly, the role of interleukin 4 (IL-4) and its receptor (IL-4R) in facilitating a pro-metastatic phenotype has been described in epithelial cancer cells [12,13,14]. IL-4R appears as a plausible candidate to attenuate metastatic tumor growth, especially since previous evidence demonstrated how antibody-mediated IL-4R neutralization lessened metastatic lung tumor burden and IL-4Rα loss reduced metastatic tumor growth [13]. Furthermore, IL-4 is able to activate the androgen receptor in conditions of low androgen levels [15]. Our finding that *IL6* rs2069832 was positively associated with a higher risk of PCa is consistent with that of Slattery et al. [16], in which rs2069832 was associated with an increased risk of breast cancer as well. Interleukin 6 (IL-6) is a pro-inflammatory cytokine expressed in PCa tissues, and its increased serum levels were observed in patients with metastatic and castration-resistant PCa (CRPC) [17,18]. Moreover, an association between IL-6 serum levels and unfavorable prognosis has also been observed in many cancers other than PCa [18,19]. *IL12RB1* rs3746190, *IL12RB2* rs2307145, and rs2229546 were associated with decreased PCa risk. Interleukin 12 (IL-12) plays a pivotal role in stimulating natural killer (NK) cell cytotoxicity, inflammation, and boosting T-helper type I cells [20]. No other studies have associated both polymorphisms with PCa, although Kundu et al. demonstrated higher serum levels of IL-12 p40 monomer in patients with PCa than in healthy individuals [21]. In particular, the IL-12 p40 monomer helped cancer cells escape cell death by suppressing IL12RB1 internalization; therefore, PCa cell apoptosis was induced after IL-12 p40 neutralization and consequent IL12RB1 internalization. With regard to *IL12RB2* rs2229546, our results are in line with another study that found an association between rs2229546 and a reduced risk of cervical squamous cell carcinoma and HPV-18-positive cervical cancer [9]. *IL17A* rs7747909 belongs to the 3′-UTR region of interleukin 17A (IL-17A) and was found associated with a higher risk of PCa. Interestingly, Bedoui et al. reported a strong association between *IL17A* polymorphisms, including rs7747909, and the development of tolerance to colon rectal cancer therapy [22]. IL-17A is a pro-inflammatory cytokine primarily produced by Th17 cells whose elevated expression has been associated with an excessive inflammatory state and tumor growth in prostate and colon cancers [23,24,25,26]. Interestingly, gene amplification of *IL17* has been reported more frequently in CRPC and neuroendocrine prostate cancer than in hormone-naïve PCa [27]. *IL13* rs20541 polymorphism, located in exon 4, appears to be strongly associated with increased serum IgE levels in children [28]. Interleukin 13 (IL-13) is primarily produced by activated Th2 cells and exhibits strong antitumor activity in PCa cell lines [29]. Moreover, rs20541 has previously been associated with cancer risk, but this association remains uncertain. In fact, in our study, *IL13* rs20541 was positively associated with the risk of PCa, whereas it was previously reported to be associated with a decreased risk of glioma [30,31]. In addition, *TMPRSS2* rs140141551 and *ACE2* rs35803318 were associated with a higher risk of PCa than *TMPRSS2* rs11701576, which was negatively associated with PCa risk. *TMPRSS2* encodes a transmembrane serine protease 2 (TMPRSS2), which is expressed by prostate epithelial cells. To date, TMPRSS2 physiological function remains unclear, albeit its expression appears to be regulated by androgens in PCa cells [32,33]. In contrast, *ACE2* encodes the angiotensin-converting enzyme (ACE2), which is aberrantly expressed in several cancers [34,35]. Compelling evidence demonstrated the strong association between ACE2 expression and tumor progression, prognosis, antitumor immunity, and immunotherapy response [36,37,38,39,40]. In fact, increased expression of ACE2 has been linked to better survival, probably due to the antitumor effects on angiogenesis and heightened immune infiltration in different cancer cohorts. Moreover, a positive correlation was also found between ACE2 expression and PD-L1, a predictive biomarker of response to immune checkpoint inhibitors. This correlation, in turn, was linked to a favorable response to immunotherapy.

*ACE2* rs35803318 is a silent polymorphism which may affect the expression of ACE2 increasing the susceptibility to PCa, especially taking into account recent studies demonstrating the effect of silent mutations on gene function and expression, splicing mechanisms, and phenotypes by affecting miRNA binding, protein folding, and mRNA stability [41,42,43,44]. Finally, the analysis displayed a significant association between *MAPT* rs2258689 and increased risk of PCa. *MAPT* encodes microtubule-associated protein tau (MAPT) whose expression is not restricted to neurons. In truth, MAPT overexpression was previously associated with lower GS, suggesting its potential as marker of PCa prognosis [45] and response to docetaxel [46]. In our analysis, we also considered whether the SNPs identified by our model were associated with the clinical information of patients with PCa, particularly the GS and PSA serum levels. GS is a system implemented to help urologists evaluate staging and predict prognosis in patients with PCa. We found that this parameter was affected by *IL2RA* rs7093069 and *TNFRSF1B* rs2275416, implying the strong impact of genetic variability within immune genes on cancer development, progression, and staging. Alterations in *IL2RA* relative expression have been reported in groups of patients with different GS [47]. On the other hand, *TNFRSF1B* rs2275416 was associated with serum PSA levels, GS, and PCa risk. *TNFRSF1B* encodes the receptor for tumor necrosis factor-α (TNFR2), which plays a crucial role in promoting tumor growth, invasion, and metastasis. TNFR2 is highly expressed in T regulatory cells (Tregs) and its presence is crucial in the TME [48]. Hence, specific TNFR2-targeted cancer therapies and TNFR2-antagonist antibodies have been tested to regulate tumor growth, showing promising results in vitro and in vivo [49,50,51]. As the presence of TNFR2 in the blood and TME has been associated with unfavorable prognosis in several types of cancer [52], it is possible that the presence of *TNFRSF1B* rs2275416 may also affect PCa prognosis, given its association with both GS and PSA levels. *SLC11A1* rs17221959 was also associated with serum PSA levels. This finding is also consistent with the results recently reported by Zhu et al. in which *SLC11A1* rs7573065 C > T was associated with increased risk of PCa and low overall survival; meanwhile, higher mRNA expression was found in PCa tissues compared to normal tissues [53]. Despite the small sample size, our study is the first to investigate these genes in PCa. For this reason, future analysis, performed on a larger scale, is necessary to further confirm our findings and explore the prognostic impact of these variants in patients diagnosed with PCa.

## 4. Materials and Methods

### 4.1. Study Population, Clinical Characteristics, and Blood Samples Collection

The ethical committee of the AOU, Sassari, approved this study. Two cohorts, comprising 47 patients with PCa and 43 HCs, were recruited. The demographic and clinical characteristics of the patients with PCa and HCs are summarized in Table 6. Patients newly diagnosed with PCa were recruited by the Urology Department of the University Hospital of Sassari, whereas HCs were recruited by the Transfusion Center (AOU, Sassari, Italy). In PCa patients, blood sample collection was performed at the time of the diagnosis. Peripheral blood samples were collected from both cohorts using K^+^-EDTA test tubes. PCa assessment was based on the clinical examination and histopathological analysis of isolated biopsies performed in all patients. 

### 4.2. Peripheral Blood Mononuclear Cell Isolation and DNA Extraction

PBMCs were isolated by Ficoll-Histopaque gradient centrifugation (Sigma-Aldrich, St. Louis, MO, USA). PBMCs were washed twice using phosphate-buffered saline (PBS 1X) and stored at −80 °C in fetal bovine serum (FBS) and dimethyl sulfoxide (DMSO) (Sigma-Aldrich, St. Louis, MO, USA). Genomic DNA extraction from PCa and HCs samples was performed using the DNeasy Blood and Tissue Kit (Qiagen, CA, USA). After thawing, cells were washed twice in PBS 1X to remove DMSO and FBS traces and then resuspended in 200 μL PBS 1X before proceeding according to the manufacturer’s instructions. The final DNA concentration of each sample was measured using Nanodrop One (Thermo Scientific, Waltham, MA, USA), and DNA quality was assessed by calculating the ratios of absorbance at A260/A280 and A260/A230.

### 4.3. Library Preparation and Genomic DNA Sequencing

Library preparation and DNA sequencing were performed at the CRS4 Research Center (Pula, Sardegna, Italy). Thirty-five genes involved in both innate and adaptive immune responses as well as PCa-associated genes were sequenced. Two AmplySeq primer pools were designed and purchased from Illumina for the gene panel analysis. The PCR protocol and reagents were adapted from QIAGEN Application Note (Qiagen 1104745 10/2016) to amplify the entire length of each gene. The PCR products were quantified using Qubit, and equimolar amounts were pooled. Gene-panel PCR pools were grouped to achieve the appropriate sequencing coverage. Libraries were obtained using Nextera DNA Flex with 100 ng of DNA and indexed with IDT for the Illumina Nextera DNA UD Indexes Primer Set (Illumina, San Diego, CA, USA). PCR products were purified with 1X AMPure XP beads (Beckman Coulter, Brea, CA, USA), and the libraries were quantified using Qubit. A loading pool consisting of 90 samples (47 PCa and 43 HCs) was diluted to 9 pM before sequencing using the MiSeq Reagent Kit v3 600-cycle (Illumina). The BaseSpace Sequence Hub (Illumina) was used to conduct demultiplexing and fastq file generation, and two distinct analyses were performed for each sample. Gene panel sequences were analyzed using an in-house pipeline. 

### 4.4. Statistical Analysis

The allele and genotype frequency estimates were calculated using (http://ihg2.helmholtzmuenchen.de/cgi-bin/hw/hwa1.pl, accessed on 1 January 2022). Fisher exact test, chi-square test, and both allelic and genotypic odds ratios (ORs) were used to compare genetic variants between PCa and HC cohorts. To test the relationship between cancer onset and nucleotide substitution for a given locus, multivariate regression analysis was carried out by applying backward selection, in which non-significant covariates were removed step by step until the final model was obtained. A GLMM implemented in the R package lme4 [54] in the R environment (version 4.1.1; R Core Team available at https://www.r-project.org/, accessed on 1 January 2022), was used to perform the analysis. Odds ratios with 97.5% confidence interval were calculated using the R-package “oddsratio” (available at https://cran.r-project.org/web/packages/oddsratio/, accessed on 1 January 2022).

## Figures and Tables

**Table 1 ijms-24-07594-t001:** Distribution of statistically significant allele and genotype frequencies in PCa patients and HCs. Values in bold represent significant results.

Gene	SNP ID	Gene Feature	Gentoypes, Alleles	PCa N (F)	HCs N (F)	OR	95% CI	*p*-Value
*IL4R*	rs2301807	intronic	AA	22 (0.512)	0 (0.000)	Ref		
			AC	1 (0.023)	0 (0.000)	0.067	0.001–4.693	Ns
			CC	20 (0.465)	32 (1.000)	0.014	0.001–0.244	<0.0001
			A	45 (0.523)	0 (0.000)	Ref		
			C	41 (0.477)	64 (1.000)	0.007	0.000–0.118	<0.0001
*IL12RB1*	rs3746190	3′ UTR	GG	22 (0.500)	12 (0.353)	Ref		
			GA	22 (0.500)	18 (0.529)	0.667	0.261–1.706	ns
			AA	0 (0.000)	4 (0.118)	0.062	0.003–1.243	0.013
			G	66 (0.750)	42 (0.618)	Ref		
			A	22 (0.250)	26 (0.382)	0.538	0.271–1.070	ns
*IL12RB2*	rs2307145	missense	GG	38 (0.950)	24 (0.800)	Ref		
			GC	2 (0.050)	5 (0.167)	2.857	0.045–1.407	ns
			CC	0 (0.000)	1 (0.033)	0.625	0.008–5.419	ns
			G	78 (0.975)	53 (0.883)	Ref		
			C	2 (0.025)	7 (0.117)	0.194	0.039–0.971	0.038
*IL6*	rs2069832	intronic	AA	1 (0.083)	2 (0.027)	Ref		
			AG	10 (0.500)	12 (0.270)	1.667	0.131–21.195	ns
			GG	26 (0.417)	10 (0.703)	5.200	0.423–63.909	ns
			A	12 (0.162)	16 (0.333)	Ref		
			G	62 (0.838)	32 (0.667)	2.583	1.092–6.114	0.028
*TMPRSS2*	rs140141551	intronic	CC	39 (0.830)	40 (0.976)	Ref		
			CA	8 (0.170)	1 (0.024)	8.205	0.980–68.710	0.024
			AA	0 (0.000)	0 (0.000)	1.025	0.020–52.950	ns
			C	86 (0.915)	81 (0.988)	Ref		
			A	8 (0.150)	1 (0.012)	7.535	0.922–61.586	0.0382
*TMPRSS2*	rs11701576	intronic	AA	38 (0.884)	18 (0.667)	Ref		
			AG	5 (0.116)	9 (0.333)	0.263	0.077–0.899	0.027
			GG	0 (0.000)	0 (0.000)	0.481	0.009–25.810	ns
			A	81 (0.942)	45 (0.833)	Ref		
			G	5 (0.058)	9 (0.167)	0.309	0.097–0.977	0.037
*ACE2*	rs35803318	synonymous	CC	43 (0.915)	34 (1.000)	Ref		
			CT	1 (0.021)	0 (0.000)	2.379	0.094–60.247	ns
			TT	3 (0.064)	0 (0.000)	5.552	0.277–131.141	ns
			C	87 (0.925)	68 (1.000)	Ref		
			T	7 (0.075)	0 (0.000)	11.743	0.659–209.216	0.022

Abbreviations: ns, not significant; Ref, reference allele; N, number of individuals; F, frequency; OR, odds ratio; CI, confidence interval.

**Table 2 ijms-24-07594-t002:** Results of multivariate analysis. The table shows the variables (potential predictors) with statistical significance with the occurrence of the illness and the summary of the coefficients found. NM denotes the number of intermediate models used to obtain the final model. Indicated values represent the results obtained for each gene individually within the multivariate analysis.

Gene	SNP ID	Estimate	Std. Error	*t* Value	Pr(>|t|)	NM
*ACE2*	rs35803318	0.5000	0.2528	1.978	0.0511 .	3
*IFNGR2*	rs9808753	0.34942	0.18920	1.847	0.0682 .	4
	rs753453319	−0.48010	0.21280	−2.256	0.0266 *	
	rs193922682	0.36132	0.13932	2.594	0.0112 *	
*IL12RB2*	rs2228420	0.66071	0.17527	3.770	0.000297 ***	4
	rs2229546	−0.38462	0.18748	−2.052	0.043225 *	
*IL13*	rs20541	6.104 × 10^−1^	1.368 × 10^−1^	4.463	2.39 × 10^−5^ ***	3
*IL17A*	rs7747909	0.23482	0.10486	2.239	0.0277 *	4
*IL18*	rs549908	0.21469	0.10577	2.030	0.0454 *	4
*IL18R1*	rs2241116	0.35721	0.13124	2.722	0.007840 **	4
	rs1420096	0.22462	0.10159	2.211	0.029647 *	
*IL2RA*	rs7093069	0.25379	0.11733	2.163	0.0333 *	3
*IL4R*	rs2301807	−0.30648	0.10317	−2.971	0.00383 **	3
*IL6*	rs2069832	0.2769	0.1072	2.582	0.011469 *	3
*IL* *7R*	rs1494555	0.26500	0.10337	2.564	0.0121 *	3
*LAG3*	rs3214312; rs397839935	0.37500	0.18269	2.053	0.0431 *	3
	rs549618226	−0.87500	0.52256	−1.674	0.0977 .	
	12:6886942 ^a^	−0.87500	0.52256	−1.674	0.0977 .	
*MAPT*	rs63750072	0.9236	0.2944	3.137	0.00234 **	4
	rs2258689	0.2375	0.1105	2.150	0.03441 *	
	rs11568305	−0.5082	0.2391	−2.126	0.03640 *	
	rs150660024	−1.6101	0.5547	−2.903	0.00471 **	
*PARK2*	6:161781121 ^a^	0.66667	0.34882	1.911	0.05927 .	3
	rs3765474	0.30128	0.10410	2.894	0.00481 **	
*SLC11A* *1*	rs17221959	0.4943	0.2919	1.693	0.094 .	3
*TMPRSS2*	rs112132031	0.3284	0.1257	2.613	0.0106 *	4
*TNFRSF1B*	rs636964	−0.52525	0.22048	−2.382	0.0194 *	4
	rs2275416	0.38889	0.17371	2.239	0.0277 *	

‘^a^’: chromosome and SNP position. Significance codes: 0 < ‘***’ < 0.001 < ‘**’ < 0.01 < ‘*’ < 0.05 < ‘.’ < 0.1.

**Table 3 ijms-24-07594-t003:** Results of the odds ratio calculation on multivariate analysis were applied to the statistically significant variables representing useful predictors. Calculations were performed on the final model with 10% increment steps across the entire predictor distribution.

Gene	SNP ID	Odds Ratio	CI Low (2.5%) *	CI High (97.5%) *
*ACE2*	rs35803318	1.05	1.05–1.10	1.05–1.00
*IFNGR2*	rs9808753	1.04	1.03–1.07	1.04–1.00
	rs753453319	0.95	0.95–0.99	0.96–0.91
	rs193922682	1.04	1.03–1.06	1.05–1.01
*IL12RB2*	rs2228420	1.07	1.04–1.10	1.10–1.04
	rs2229546	0.96	0.93–0.99	1.00–0.94
*IL13*	rs20541	1.06	1.04–1.07	1.09–1.06
*IL17A*	rs7747909	1.02	1.01–1.04	1.04–1.01
*IL18*	rs549908	1.02	1.01–1.04	1.04–1.01
*IL18R1*	rs2241116	1.04	1.03–1.06	1.04–1.01
	rs1420096	1.01	1.01–1.04	1.04–1.01
*IL2RA*	rs7093069	1.03	1.02–1.05	1.04–1.01
*IL4R*	rs2301807	0.97	0.96–0.98	0.98–0.96
*IL6*	rs2069832	1.03	1.01–1.04	1.05–1.02
*IL* *7R*	rs1494555	1.03	1.01–1.04	1.04–1.01
*LAG3*	rs3214312; rs397839935	1.04	1.03–1.07	1.05–1.00
	rs549618226	0.92	0.95–1.02	0.88–0.83
	12:6886942 ^a^	0.92	0.95–1.02	0.88–0.83
*MAPT*	rs63750072	1.10	1.08–1.16	1.11–1.04
	rs2258689	1.02	1.01–1.04	1.04–1.01
	rs11568305	0.95	0.94–1.00	0.97–0.91
	rs150660024	0.85	0.89–0.95	0.82–0.76
*PARK2*	6:161781121 ^a^	1.07	1.08–1.15	1.06–1.00
	rs3765474	1.03	1.01–1.04	1.05–1.02
*SLC11A* *1*	rs17221959	1.05	1.06–1.11	1.05–0.99
*TMPRSS2*	rs112132031	1.03	1.01–1.04	1.06–1.02
*TNFRSF1B*	rs636964	0.95	0.94–0.99	0.96–0.91
	rs2275416	1.04	1.02–1.07	1.06–1.01

* Range of odds ratios within the 10 steps that split the predictor distribution. ‘^a^’: chromosome and SNP position.

**Table 4 ijms-24-07594-t004:** Odds ratio calculations on multivariate analysis were applied to statistically significant variables identified by GLMM model.

Gene	SNP ID	OR	CI_Low (2.5%)	CI_High (97.5%)	Pr(>|t|)
*IL12RB2*	rs2228420	1.687	1.282	2.219	0.000347 ***
	rs2229546	0.606	0.447	0.822	0.001876 **
*IL13*	rs20541	1.850	1.369	2.500	0.000138 ***
*IL17A*	rs7747909	1.214	1.040	1.418	0.015979 *
*IL4R*	rs2301807	0.653	0.555	0.768	1.87 × 10^−6^ ***
*MAPT*	rs2258689	1.243	1.052	1.468	0.012260 *
*TNFRSF1B*	rs636964	0.678	0.476	0.965	0.034034 *
	rs2275416	1.377	1.042	1.820	0.027425 *

Statistical significance: 0 < ‘***’ < 0.001 < ‘**’ < 0.01< ‘*’ < 0.05.

**Table 5 ijms-24-07594-t005:** The GLMM analysis between the clinical characteristics of patients with PCa and the predictors is shown in Table 3.

Clinical Variable	Gene	Predictors	Estimate	Std. Error	t Value	Pr(>|t|)
GS	*IL2RA*	rs7093069	0.5864	0.2866	2.046	0.0472 *
	*TNFRSF1B*	rs2275416	0.7624	0.3618	2.107	0.0413 *
PSA	*SLC11A1*	rs17221959	32.799	11.915	2.753	0.00869 **
	*TNFRSF1B*	rs636964	−33.887	12.208	−2.776	0.00819 **
		rs2275416	23.014	7.629	3.017	0.00433 **

Abbreviations: GS, Gleason score; PSA, prostate-specific antigen. Significance codes: 0.001 < ‘**’ < 0.01 < ‘*’ < 0.05.

**Table 6 ijms-24-07594-t006:** Clinical and demographic information of the PCa and HCs cohorts.

	PCa (n = 47)	HCs (n = 43)
Age (mean ± SD)	70.6 ± 8.0	49.7 ± 14.9
Serum PSA (ng/mL, mean ± SD)		
≤4 ng/mL	7	
>4 ng/mL	40	
GS		
GS = 6	20	
GS = 7	16	
GS ≥ 8	8	

Abbreviations: GS, Gleason score; PSA, prostate-specific antigen.

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
