# Peer review of "A Multigene-Panel Study Identifies Single Nucleotide Polymorphisms Associated with Prostate Cancer Risk"

_ijms, 2023, doi:10.3390/ijms24087594_

Round 1

Reviewer 1 Report

In this manuscript, Manca et al. provide evidence of the tight interplay between tumor cells and microenvironment, giving insight into the prognostic impact of genetic variants in prostate cancer. 

Minor concerns:

- 52 / 226 : I suggest to specify that patients are new diagnosed (if so) or add in the table the specific disease timepoint when the samples were collected. 

-124:  I suggest to better explain the associations with the Gleason score, are the polymorphism associated with Higher Gleason score? please specify

- 144 : if possible could be worth to perform the flow characterization of CD4/Cd8 subpopulations in patients and healthy donors, to possibly correlate with the specific polymorphisms signature.

Author Response

Minor concerns:

- 52 / 226 : I suggest to specify that patients are new diagnosed (if so) or add in the table the specific disease timepoint when the samples were collected.

Response: All the PCa patients are newly diagnosed and the PBMCs were collected at the moment of diagnosis. We added such information related to PCa patients in the paragraph 4.1, (line 241 and 243).

-124:  I suggest to better explain the associations with the Gleason score, are the polymorphism associated with Higher Gleason score? please specify

Response: Yes, IL2RA rs7093069 and TNFRSF1B rs2275416 are strongly associated with higher values of GS. Accordingly, they represent good predictors for severity of the pathology. This point has been better explained in the revised version of the manuscript (lines 125-128 and 130-132).

- 144 : if possible could be worth to perform the flow characterization of CD4/Cd8 subpopulations in patients and healthy donors, to possibly correlate with the specific polymorphisms signature.

Response: This is certainly an interesting suggestion and is worth to investigate. However, at this point it is not possible to perform the characterization of CD4/CD8 population using flow cytometry, it will be done in future studies.

Reviewer 2 Report

This interesting paper is full of numbers and, probably due to this, there are some inconsistencies. Please read the paper carefully to be certain that all the figures are complete and consistent.

For example:

At page 2 line 67 the first CI is lacking

In Table 1, relative frequencies for IL12RB1 are lacking

In table 2 rs753453319 as well as rs2301807 beta estimates should be negative accordingly to t value

Moreover, at page 2, line 69, line 74, line 78 as well as in Table 1, some 95% CIs include the value 1, how can they be significant?

In table 2. Results of multivariate analysis for each gene individually are reported?  What does it mean? If the analysis is 'individually' it is not multivariable.

T-Fisher is an unusual notation, please explain.

Reference 18 e 19 are the same

My main concern regards age. Healthy control is really younger than patients, age should be included in the model, to adjust coefficient estimation. 

Author Response

This interesting paper is full of numbers and, probably due to this, there are some inconsistencies. Please read the paper carefully to be certain that all the figures are complete and consistent.

Answer: We thank the reviewer for the appreciation of the work and apologize for the inconsistencies which were corrected.

For example:

At page 2 line 67 the first CI is lacking: 

Response: We apologize and added the missing 95% CI (line 67).

In Table 1, relative frequencies for IL12RB1 are lacking.

Response: We added the missing relative frequencies (Table 1).

In table 2 rs753453319 as well as rs2301807 beta estimates should be negative accordingly to t value

Response: That were typos. Values in Table 2 have been corrected.

Moreover, at page 2, line 69, line 74, line 78 as well as in Table 1, some 95% CIs include the value 1, how can they be significant?

Response: The analysis of ORs and 95% CI were performed using the online tool (http://ihg2.helmholtzmuenchen.de/cgi-bin/hw/hwa1.pl). The values displayed in Table 1 are those calculated by the tool.

In table 2. Results of multivariate analysis for each gene individually are reported?  What does it mean? If the analysis is 'individually' it is not multivariable.

Response: The performed analysis is a multivariate analysis. We indicated in the Table also results obtained individually for each variant within the multivariate analysis. In our opinion, this point is very important because it gives the magnitude of information and the weight of each variable within the multivariables pool. This point has been now better explained in the revised version of the manuscript.

T-Fisher is an unusual notation, please explain.

Response: The test used to perform the analysis is a Fisher exact test and we changed the test name in the description of the statistical methods used (line 278).

Reference 18 e 19 are the same:

Response: We changed and corrected the list of references.

My main concern regards age. Healthy control is really younger than patients, age should be included in the model, to adjust coefficient estimation.

Response: Age has been included in the model, but as variable, age was not statistically significant. Indeed, our model that has been developed by using as backward selection (in which non-significant covariates were removed step by step until the final model was obtained) excluded age.

Reviewer 3 Report

Pitifully the first mention that I have to make is not related to the quality of the work, but to the fact that it is not a “Review” but rather a “Research” article. Thus, it cannot be evaluated as a Review. Please resubmit your manuscript to the correct section of the journal or ask the editors to move it for you (in case this were somebody’s else mistake).

Regarding the research presented on the work, the manuscript described the investigation of the association between a set of gene single nucleotide polymorphisms (SNPs) and prostate cancer (PCa) risk. Although important evidence is shown on possible associations between SNP and cancer development, the paper have essential flaws/problems which should be corrected/solved in order to be Published on IJMS:

1) The title mention the word “variants” which indeed is associated with single nucleotide variants (SNV), a genetic feature that is represented in less than 1% of the population, in contrast to SNP, that are present in at least 1% of the population. Please use a synonym or directly single nucleotide polymorphism in the title instead of the word “variants’ (for the sake of clarity).

2)The words “single nucleotide polymorphism” did not appear in the abstract, which makes obscure the nature of your analysis. Please incorporate it in the abstract.

3) The final sentence in the abstract (lines 25- 26: “Our results corroborate the hypothesis that a systemic imbalance of the major players in the immune system can affect susceptibility to prostate cancer”.) is overstating your results since you did not corroborate the hypothesis of immune imbalance (you do not know if the expression of the gene products or the T helper predominant response in order to state this). Please tone down.

4) In order to be published this work needs to add supplementary material giving extensive detail of a) the statistical methods applied , b) the previous works that were used to design the panel of SNPs, c) how and why some genes were used and others that could be involved in PCa progression were neglected and d) the meaning of Gleason scores and these plotted vs SNPs in a graph.

5) Although it was previously concluded that: “The evidence on currently available SNP panels is insufficient to assess analytic validity, and at best the panels assessed would add a small and clinically unimportant improvement to factors such as age and family history in risk stratification (clinical validity).No evidence on the clinical utility of current panels is available” [1], I’ll assume that you can add clinical data other than GS, and which is more important, protein expression and immunological outcomes. Therefore, since you are expected (by law) to properly store the patients samples used for this study, I will ask you to study in these 90 patients samples (or at least in the most representatives) the relevant cytokine expression panels (can be done by ELISA/flow cytometry with beads) and T helper activation and profiling data, with the corresponding analysis, in order to be published in this manuscript. The usefulness of the whole work could be greatly improved if you add the mentioned analysis.

6) In lines 144 – 145 authors stated that: “...in T-helper cytokine 144 genes (IL13 rs20541 and IL4R rs2301807) suggests the hypothesis of an imbalance in the T-145 helper response of PCa patients.”. which is not the whole truth, IL13/4, either protein or receptor are not exclusively associated to T helper, and there is no way that this could suggest a T helper imbalance if you do not have protein (cytokine) expression data or T helper profiling in those patients. Please rephrase in a descriptive and less speculative manner.

7)In lines 155 – 156 you claimed that IL- 6 is also an anti-inflammatory cytokine . Although there are evidence of such thing, allegedly occurring during exercise in muscle (in which particular case they call it myokine), IL-6 is pro-inflammatory cytokine in most cases, in most pathologies and also, specifically, in PCa progression. Please erase the confusing anti-inflammatory part or rephrase the sentence to keep the focus on your topic.

8)In lines 190 to 192 the authors stated that: “ Compelling evidence demonstrated the strong association between ACE2 expression and tumor progression, prognosis, anti-tumor immunity and immunotherapy response” … which is unclear if associated with a bad or good prognosis, poor or strong immunity? poor or adequate immunotherapy response?

Please rephrase and clarify.

9)Last but not least: In 2011 the Stockholm-1 cohort study published the result of the evaluation of 5241 patients (2135 positive for PCa) with a panel of 35 SNP also [2]. That amount of work done 12 years ago when the technology were not as advanced as nowadays, were carried out by 8 (eight) authors. My question for you is hoy you can justify that 12 (twelve) years ago for the analysis of 5241 patients you need just 8 (eight) authors and in your paper for the analysis of 90 (ninety) patients you need to be 12 (twelve) authors? I will not go in details on how you justify to be such a crowd, simply I will kindly ask you to reduce the unnecessary/unjustified author names on your list, reminding you that “Ghost authorship” is also regarded as scientific misconduct. Please erase the names of not significant contributions for this work.

If you kindly comply with the needed corrections and the suggested analysis on cytokines/T helper the paper could be published without further editing.

References:

1. Little J, Wilson B, Carter R, Walker K, Santaguida P, Tomiak E, Beyene J, Usman Ali M, Raina P. Multigene panels in prostate cancer risk assessment: a systematic review. Genet Med. 2016 Jun;18(6):535-44. doi: 10.1038/gim.2015.125. Epub 2015 Oct 1. PMID: 26426883.

2. Aly M, Wiklund F, Xu J, Isaacs WB, Eklund M, D'Amato M, Adolfsson J, Grönberg H. Polygenic risk score improves prostate cancer risk prediction: results from the Stockholm-1 cohort study. Eur Urol. 2011 Jul;60(1):21-8. doi: 10.1016/j.eururo.2011.01.017. Epub 2011 Jan 18. PMID: 21295399; PMCID: PMC4417350.

Author Response

Pitifully the first mention that I have to make is not related to the quality of the work, but to the fact that it is not a “Review” but rather a “Research” article. Thus, it cannot be evaluated as a Review. Please resubmit your manuscript to the correct section of the journal or ask the editors to move it for you (in case this were somebody’s else mistake).

Answer: Obviously it has be changed Review into Article, we do not understand at which point was inserted the error and will ask the editors to correct it.

Regarding the research presented on the work, the manuscript described the investigation of the association between a set of gene single nucleotide polymorphisms (SNPs) and prostate cancer (PCa) risk. Although important evidence is shown on possible associations between SNP and cancer development, the paper have essential flaws/problems which should be corrected/solved in order to be Published on IJMS:

1) The title mention the word “variants” which indeed is associated with single nucleotide variants (SNV), a genetic feature that is represented in less than 1% of the population, in contrast to SNP, that are present in at least 1% of the population. Please use a synonym or directly single nucleotide polymorphism in the title instead of the word “variants’ (for the sake of clarity).

Response: we substituted “variants” with SNPs in the title.

2) The words “single nucleotide polymorphism” did not appear in the abstract, which makes obscure the nature of your analysis. Please incorporate it in the abstract.

Response: we incorporated “single nucleotide polymorphism” in the abstract.

3) The final sentence in the abstract (lines 25- 26: “Our results corroborate the hypothesis that a systemic imbalance of the major players in the immune system can affect susceptibility to prostate cancer”.) is overstating your results since you did not corroborate the hypothesis of immune imbalance (you do not know if the expression of the gene products or the T helper predominant response in order to state this). Please tone down.

Response: We toned down our statement on the results obtained in the present study (line 25-27).

4) In order to be published this work needs to add supplementary material giving extensive detail of a) the statistical methods applied , b) the previous works that were used to design the panel of SNPs, c) how and why some genes were used and others that could be involved in PCa progression were neglected and d) the meaning of Gleason scores and these plotted vs SNPs in a graph.

Response: In the analysis we included also age, cTNM classification and body mass index values. However, no significant association was found between the observed SNPs and these clinical characteristics. For this reason, we decided to report only the SNPs significantly associated with Gleason score and PSA.

5) Although it was previously concluded that: “The evidence on currently available SNP panels is insufficient to assess analytic validity, and at best the panels assessed would add a small and clinically unimportant improvement to factors such as age and family history in risk stratification (clinical validity). No evidence on the clinical utility of current panels is available” [1], I’ll assume that you can add clinical data other than GS, and which is more important, protein expression and immunological outcomes. Therefore, since you are expected (by law) to properly store the patients samples used for this study, I will ask you to study in these 90 patients samples (or at least in the most representatives) the relevant cytokine expression panels (can be done by ELISA/flow cytometry with beads) and T helper activation and profiling data, with the corresponding analysis, in order to be published in this manuscript. The usefulness of the whole work could be greatly improved if you add the mentioned analysis.

Response: Indeed the patients samples were properly store, but not for all of them a significative amount of sample was remained in order to proceed with the suggested investigations: “Relevant cytokine expression panels and T helper activation and profiling data”. These analyses will be done in future studies in order to associate the SNP panels to analytic validity (as we previously concluded), and it is outside of our present research founds at the moment.

6) In lines 144 – 145 authors stated that: “...in T-helper cytokine genes (IL13 rs20541 and IL4R rs2301807) suggests the hypothesis of an imbalance in the T-helper response of PCa patients.”. which is not the whole truth, IL13/4, either protein or receptor are not exclusively associated to T helper, and there is no way that this could suggest a T helper imbalance if you do not have protein (cytokine) expression data or T helper profiling in those patients. Please rephrase in a descriptive and less speculative manner.

Response: We rephrased our statement as suggested (line 150-152).

7) In lines 155 – 156 you claimed that IL- 6 is also an anti-inflammatory cytokine . Although there are evidence of such thing, allegedly occurring during exercise in muscle (in which particular case they call it myokine), IL-6 is pro-inflammatory cytokine in most cases, in most pathologies and also, specifically, in PCa progression. Please erase the confusing anti-inflammatory part or rephrase the sentence to keep the focus on your topic.

Response: We rephrased the sentence about IL-6 as suggested (line 162-164).

8) In lines 190 to 192 the authors stated that: “ Compelling evidence demonstrated the strong association between ACE2 expression and tumor progression, prognosis, anti-tumor immunity and immunotherapy response” … which is unclear if associated with a bad or good prognosis, poor or strong immunity? poor or adequate immunotherapy response?

Please rephrase and clarify.

Response: the association between ACE2 expression and cancer regards the fact that ACE2 downregulation was found correlated with worse prognosis and survival. Other studies also describe an antitumor effects of ACE2 expression, among which we find inhibition of angiogenesis and stimulation of immune cell infiltration. Finally, ACE2 expression positively correlated with PD-L1, which is a predictive biomarker of response to immune checkpoint inhibitors. We rephrased and clarified the role of ACE2 and its association with cancer prognosis and immunotherapy (line 196-203).

9)Last but not least: In 2011 the Stockholm-1 cohort study published the result of the evaluation of 5241 patients (2135 positive for PCa) with a panel of 35 SNP also [2]. That amount of work done 12 years ago when the technology were not as advanced as nowadays, were carried out by 8 (eight) authors. My question for you is hoy you can justify that 12 (twelve) years ago for the analysis of 5241 patients you need just 8 (eight) authors and in your paper for the analysis of 90 (ninety) patients you need to be 12 (twelve) authors? I will not go in details on how you justify to be such a crowd, simply I will kindly ask you to reduce the unnecessary/unjustified author names on your list, reminding you that “Ghost authorship” is also regarded as scientific misconduct. Please erase the names of not significant contributions for this work.

If you kindly comply with the needed corrections and the suggested analysis on cytokines/T helper the paper could be published without further editing.

Response: I agree that the reviewer may have some perplexity about the number of authors in the present manuscript in comparison with the 2011 Stockholm study. Indeed, all listed authors have given a real contribute to the work, that has not been done in an easy setting, and a lot of collaboration has been done in order to finalize the manuscript. Anyway we accept the invite and deleted the least contributing author (Hajra Ashraf) to the paper.

References:

  1. Little J, Wilson B, Carter R, Walker K, Santaguida P, Tomiak E, Beyene J, Usman Ali M, Raina P. Multigene panels in prostate cancer risk assessment: a systematic review. Genet Med. 2016 Jun;18(6):535-44. doi: 10.1038/gim.2015.125. Epub 2015 Oct 1. PMID: 26426883.

  1. Aly M, Wiklund F, Xu J, Isaacs WB, Eklund M, D'Amato M, Adolfsson J, Grönberg H. Polygenic risk score improves prostate cancer risk prediction: results from the Stockholm-1 cohort study. Eur Urol. 2011 Jul;60(1):21-8. doi: 10.1016/j.eururo.2011.01.017. Epub 2011 Jan 18. PMID: 21295399; PMCID: PMC4417350.

Round 2

Reviewer 2 Report

You addressed all my issues. Thank you!

Author Response

Thanks to the reviewer for the appreciation  of the manuscript

Reviewer 3 Report

Although minor style corrections had been made, not substantial changes or additional experiments were done. 

Author Response

All substantial changes were made according to the editor comments